# Diminishing Non-important Gradients for Training Dynamic Early-Exiting Networks

## Abstract

Early-exiting is an effective mechanism to improve computation efficiency. By adding classifiers to intermediate layers of deep learning networks, early exiting networks can terminate the inference early for easy samples, thus reducing the average inference time. Gradient conflicts between different classifiers are a key challenge in training early-exiting networks. However, current state-of-the-art methods focus solely on the trade-off between gradients, without evaluating whether these gradients are actually necessary. To mitigate this issue, we propose a novel adaptive damping training strategy that adaptively diminishes non-important gradients during the training process based on data samples and classifiers. By adding a damping neuron to the last fully connected layer of each classifier and using our proposed damping loss, our approach effectively reduces gradients that are unlikely to be beneficial. Moreover, we propose power-sqrt loss to concentrate the gradients of damping neurons on classifiers that exhibit relatively better training performance. Experiments on CIFAR and ImageNet demonstrate our proposed method gains significant accuracy improvement for all classifiers with negligible computation increases.

## 1 Introduction

Although deep neural networks have achieved remarkable success across various tasks Krizhevsky et al. (2012); Simonyan & Zisserman (2014); He et al. (2016); Huang et al. (2019), their high computational costs limit their application on resource-constrained devices. Many efforts have been made to improve the inference efficiency of deep neural networks such as network pruning LeCun et al. (1989); Yang et al. (2021), weight quantization Hubara et al. (2016); Han et al. (2015), and lightweight network architecture design Howard et al. (2017); Zhang et al. (2018); Sandler et al. (2018). While these efficient models achieve competitive accuracy, many challenging data samples still demand the use of larger networks Huang et al. (2017); Lin et al. (2017). By exploiting this fact, dynamic networks Han et al. (2021), which perform a data-dependent inference procedure by dynamically adjusting the network structure, have attracted considerable research interest. As a typical dynamic network, early-exiting attaches multiple intermediate classifiers (early exits) to the network. In the inference stage, early-exiting networks adaptively terminate inference when an early exit satisfies the predefined exiting criterion such as the confidence score of softmax Huang et al. (2017) or according to a learned policy Chen et al. (2020).

Unlike conventional deep neural networks, early-exiting networks have multiple exits that share parameters. This shared structure causes interference among exits. Gradients from different exits often conflict during training Sun et al. (2022). The current state-of-the-art methods of training early exiting networks adopts a meta-learning approach, where a meta-network is used to learn the weights of individual gradients during the training of early exiting networks Han et al. (2022); Sun et al. (2022). This approach belongs to the category of linear scalarization Hu et al. (2024), a mainstream method in multi-task learning. It addresses the trade-off between gradients from different tasks during the training of early exiting networks, thereby mitigating the issue of gradient conflicts.

Despite the advances, current meta-learning methods Han et al. (2022); Sun et al. (2022) do not fully consider whether the gradients from each classifier are actually necessary for its performance. **We observe that during training, samples with larger softmax values are more likely to produce unnecessary gradients.** Conventional overfitting mitigation methods cannot jointly account for the

multiple classifiers involved in training early-exiting networks. For example, in early stopping, each classifier in an early-exiting network reaches its optimal stopping point at a different time. For label smoothing, different classifiers in an early-exiting network require different smoothing strengths. Thus, an important question arises when training early-exiting networks: **How can we suppress unnecessary gradients while jointly considering the training states of all classifiers?**

To address this issue, we propose a novel training strategy that adaptively diminishes gradients based on the classifier's performance with the current data sample. As illustrated in Figure 1, the classifier diminishes the gradient when it already performs well on a given sample, as indicated by a sufficiently high softmax score. Specifically, when the classifier already achieves a high softmax score on the current data sample, our damping neuron gets assigned a higher gradient, effectively preventing unnecessary gradients. Conversely, when the classifier's performance is suboptimal and the softmax score is low, we introduce only a minimal gradient to the damping neuron, limiting its influence at this stage. Moreover, we introduce the *power-sqrt* loss function,

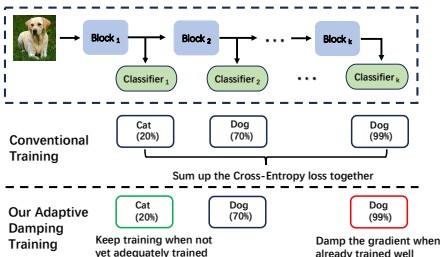

Figure 1: **Our adaptive damping training strategy.** During training, different classifiers evolve at different rates. Instead of summing the cross-entropy losses, our method applies adaptive gradient damping, reducing the influence of already well-trained classifiers.

which refines the distribution of gradients in damping neurons by concentrating them on the classifiers that exhibits superior performance relative to others. This strategy enables a more effective damping mechanism by jointly considering the status of all classifiers, thereby further improving the training process across the network. Additionally, we leverage the values of the damping neuron to assign dynamic weights to different classifiers, demonstrating the compatibility of our method with the linear scalarization approach. Our main contributions are summarized as follows:

1. We introduce a novel adaptive damping mechanism that dynamically reduces unnecessary gradients during training, improving overall performance;

2. We further propose the *power-sqrt* loss, which jointly considers the training status of all classifiers to more effectively determine the appropriate damping gradient magnitude;

3. We further demonstrate the compatibility of our method with the current linear scalarization approaches;

4. We perform extensive experiments on CIFAR Krizhevsky et al. (2009) and ImageNet Deng et al. (2009) datasets demonstrating the superiority of our method, which achieves a significant improvement in accuracy with negligible complexity increases.

## 2 RELATED WORK

**Dynamic early exiting networks.** Early exiting exemplifies a dynamic neural network architecture that enables easy samples to be processed and output by intermediate classifiers. This technique has attracted widespread attention across domains such as computer vision Huang et al. (2017); Kouris et al. (2022); Yang et al. (2023); Yu et al. (2024); Niu et al. (2024); Jiang et al. (2024); Wang et al. (2021); Yang et al. (2020), natural language processing Bajpai & Hanawal (2024); Zhou et al. (2020); Elbayad et al. (2020); Xin et al. (2021); Mangrulkar et al. (2022); Schuster et al. (2022), and multimodal tasks Tang et al. (2023); Fei et al. (2022); Yue et al. (2024).

Training dynamic early-exiting models presents unique challenges due to gradient conflicts among different exits, which compete to update shared parameters Sun et al. (2022). DFS Gong et al. (2024) mitigates gradient conflicts through feature partitioning. Meta-learning techniques Sun et al. (2022); Han et al. (2022) have also been explored to dynamically weight gradients from different exits, thereby reducing the gradients conflict. However, they did not consider whether the gradients provided by different classifiers actually benefit the model. This paper focuses on identifying and utilizing gradients that are truly beneficial to the model's performance.

**Multi-task learning.** Compared to training early-exiting networks, multi-task learning has received more attention. The primary challenge in multi-task learning is managing gradient conflicts between

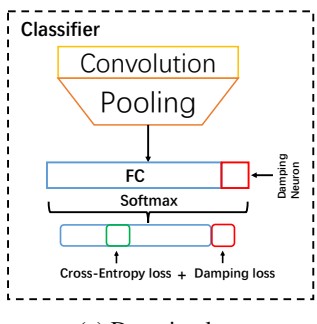
(a) Damping loss

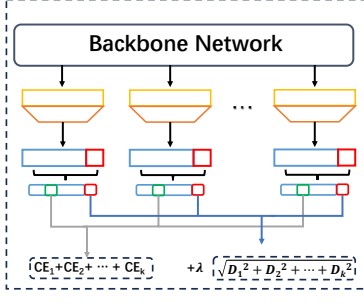
(b) Power-sqrt loss

Figure 2: **The damping neuron and power-sqrt loss.** (a) We introduced an additional neuron (red square) to the fully connected layer of the classifier, which does not correspond to any class. By assigning a small gradient to this neuron, we enable adaptive damping of the gradients of the Cross-Entropy part. (b) When jointly training classifiers, we aggregate the Cross-Entropy losses of all classifiers. For the damping neurons, we first square their values, sum them, and then take the square root (rounded rectangle in red). This approach concentrates the gradients of the damping neurons more on classifiers that perform relatively better for the current data sample, thereby enhancing the joint training of these classifiers.

different tasks Yu et al. (2020), a problem that is also prevalent in early-exiting networks. Techniques from multi-task learning, such as knowledge distillation Xu et al. (2023); Ghiasi et al. (2021), feature partitioning Ding et al. (2023) and gradient selection Liu et al. (2021), have also been applied to the training of early-exiting models Li et al. (2019); Phuong & Lampert (2019); Sun et al. (2022); Han et al. (2022); Gong et al. (2024); Addad et al. (2025). The state-of-the-art methods for training early-exiting networks use meta-learning to adaptively weight the losses of different classifiers. These approaches essentially follow the linear scalarization framework from multi-task learning Xin et al. (2022); Hu et al. (2024), where all loss terms are combined using a weighted sum. In contrast to previous methods that focus on resolving gradient conflicts, our work examines whether all gradients are necessary in the first place. As such, our approach is complementary to conflict-resolution techniques and can be integrated with them. Our approach can be integrated with linear scalarization, but in contrast to meta-learning strategies, it offers a simpler alternative by directly deriving gradient weights from damping neuron values, thereby avoiding the extra backpropagation steps and reducing training overhead.

**Overfitting.** A key challenge in neural networks is inadequate generalization, particularly in adversarial learning Kim et al. (2021) and generative models Loaiza-Ganem et al. (2022). Many studies address overfitting using regularization techniques such as dropout Srivastava et al. (2014) and label smoothing Szegedy et al. (2015). Early stopping Prechelt (2002) is a widely used technique to prevent overfitting, typically identifying the best epoch on the validation set and stopping training before overfitting occurs. However, in dynamic early-exiting networks, traditional methods for mitigating overfitting are not directly applicable, as classifiers at different exits are trained to different extents. Our work specifically targets early-exiting architectures, addressing unnecessary gradients while jointly considering the distinct roles and training states of intermediate classifiers.

## 3 METHODOLOGY

In this section, we first present conventional early exiting networks and the current training methodologies. Then, we provide a detailed explanation of our proposed approach.

### 3.1 PRELIMINARIES

**Early Exiting Networks.** Contrasting with standard deep learning models, $K$-exit early exiting networks integrate $K - 1$ classifiers at various layers within the original deep learning architecture (see Fig. 1). The prediction for the $i$-th input by classifier $f^{(k)}$ is denoted as $p_i^{(k)} = f^{(k)}(\mathbf{x}_i, \theta^{(k)})$.

where $\mathbf{x}_i$ indicates input data sample and $\theta^{(k)}$ the parameters of $k$-th classifier. Note that these sub-networks share a portion of their parameters.

Early exiting networks enable dynamic inference. They use an exit mechanism based on a confidence score. This score is typically the maximum output of the classifier's softmax result Huang et al. (2017); Yang et al. (2020). When a classifier's score reaches a predefined threshold, the model stops the inference process at this classifier, saving computational resources.

**Training Strategies of Early Exiting Networks.** The conventional training strategy for early exiting networks aggregates the losses from all classifiers, with all classifiers trained simultaneously from beginning to end Huang et al. (2017); Yang et al. (2020). The total loss is computed as $\mathcal{L} = \frac{1}{S} \sum_{k=1}^{K} \sum_{i=1}^{S} \mathcal{L}_i^{(k)}$, where $\mathcal{L}_i^{(k)} = \text{CE}(f^{(k)}(\mathbf{x}_i, \theta^{(k)}), y_i)$ denotes the cross-entropy loss for the $k$-th classifier on the $i$-th data sample. Here, $K$ is the number of classifiers, $S$ the number of training samples, and $y_i$ the ground truth label.

Training early-exiting networks often encounters gradient conflicts among classifiers. Existing methods focus on resolving these conflicts by balancing different gradients but overlook a critical question: Are all gradients necessary? Most approaches assume that every gradient should be incorporated into the trade-off, without considering whether these gradients are necessary.

## 3.2 Should All Gradients Be Considered in the Trade-off?

A recent study Wei et al. (2022) suggests that cross-entropy loss drives the softmax value to continue increasing even when it has already reached a high value, which may not always benefit model performance.

To investigate this effect, we conducted experiments on the MSDNet architecture in CI-FAR100 dataset, training only the deepest classifier to assess whether the gradients it receives improve its own performance. We introduced a threshold based on the softmax value corresponding to the correct label during training. Specifically, we discarded gradients when the softmax value exceeded a given threshold. For

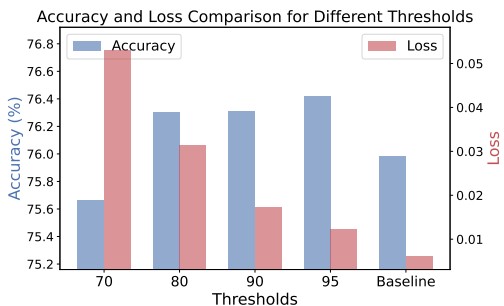

Figure 3: The relationship between softmax values and gradient necessity. As softmax increases, gradients are more likely to be unnecessary or even harmful.

instance, a threshold of 90 means that only gradients where the correct label's softmax value is below 90% are considered.

We recorded the final converged accuracy and loss under different thresholds. Our experimental results show that as more gradients are included, the final converged loss decreases. Maintaining a higher loss allows for more optimization flexibility, which provides additional capacity for other classifiers in early-exiting architectures. Despite this, the performance does not continue to improve significantly. Specifically, we observe that gradients corresponding to softmax values between 70% and 80% significantly improve model performance, while those in the 80%–95% range contribute little, and those in the 95%–100% range even degrade performance. We observe that as the softmax confidence increases, the corresponding training gradients are increasingly likely to be unnecessary.

While including more gradients reduces the final loss, accuracy sometimes decreases instead, suggesting an overfitting phenomenon. When softmax values are already high, cross-entropy loss continues to push them further, potentially leading the model to learn features that are less generalizable and primarily beneficial to a small subset of data samples.

## 3.3 Damping loss

In order to tackle the aforementioned issues, we propose a dynamic damping mechanism to diminish these non-beneficial gradients for training early-exiting.

We have modified the Cross-Entropy loss by adding our novel damping neuron, which does not represent any specific class in the classification task as shown in Figure 2a. The Softmax function

is jointly applied to the outputs from the original fully connected layer and our newly introduced damping neuron:

$$F_\theta(x)[j] = \frac{\exp\left(f_\theta(x)[j]\right)}{\sum_{n=1}^{N+1} \exp\left(f_\theta(x)[n]\right)}, \tag{1}$$

where $N$ is the number of classes in the classification task, $f_\theta(x)[j]$ denotes the output of the neurons from both the fully connected layer and our added damping neuron, and $F_\theta(x)[j]$ is the output of the Softmax function. We add an additional neuron, increasing the total number of neurons to $N + 1$. Unlike traditional Cross-Entropy loss, which solely focuses on increasing the neuron value of the correct label, our damping loss additionally provides a small gradient to the $N + 1$ neuron, which damps the gradient from the cross-entropy part:

$$\min_\theta \mathbb{E}_{t(x)}\left[-\log F_\theta(x)[y]\right] + \lambda \mathbb{E}_{t(x)}\left[-\log F_\theta(x)[N+1]\right], \tag{2}$$

where $t(x)$ denotes the training dataset, $y$ presents the correct label, and $\lambda$ is a hyperparameter, typically assigned a small value to ensure that the majority of our loss remains focused on the classification task. The first term of our loss function is the Cross-Entropy loss, while the second term is designed to encourage an increase in the $N + 1$-th neuron for any data sample. Additionally, this gradient counteracts the effect of the gradient from the Cross-Entropy term, thereby dampening its influence. Our method introduces only one additional neuron to the fully connected layer, thereby rendering the extra computational demand negligible.

Our in-depth theoretical analysis demonstrates several advantages of our damping loss:

1. The damping component of our loss generates stronger inhibitory gradients as the softmax value increases, effectively suppressing unnecessary gradients that are more likely to appear at higher softmax values.

2. Unlike Cross-Entropy loss, our damping loss does not continuously encourage the softmax value of the correct label to approach 1, preventing the generation of potentially unnecessary gradients.

3. Our damping loss does not perpetually promote the increase of the damping neuron's softmax value, preventing adverse effects on the model's performance.

We provide propositions with the proof sketch below, with the detailed proofs included in the Appendix B.1.

**Proposition 1.** The gradient of the damping component with respect to the neuron corresponding to the correct label $\frac{\partial -\log F_\theta(x)[N+1]}{\partial f_\theta(x)[y]}$ is proportional to its softmax value $F_\theta(x)[y]$. When the neuron $F_\theta(x)[y]$ achieves $\frac{1}{1+\lambda}$, the gradient from our damping loss $g \geq 0$.

**Proof sketch.** We demonstrate our proposition by analyzing the gradients generated by our damping loss for each neuron in the fully connected layer. Because our damping loss comprises two components, each neuron in the fully connected layer is influenced by gradients from both the Cross-Entropy component and the damping component.

For the neuron corresponding to the correct label, the gradient from the damping component of our damping loss is $\lambda F_\theta(x)[y]$, which is positive and thus encourages a reduction in the neuron value. This gradient is proportional to the softmax value of the neuron corresponding to the correct label, meaning that for larger softmax values, where gradients are more likely to be unnecessary, it generates a stronger opposing effect to suppress further increases.

Moreover, the gradient from the cross-entropy component of the loss is $F_\theta(x)[y] - 1$. Thus, the total gradient for the neuron corresponding to the correct label is $(\lambda + 1)F_\theta(x)[y] - 1$. Our damping loss does not continuously encourage the softmax value of the correct label to grow. Once it reaches $\frac{1}{1+\lambda}$, the overall gradient becomes positive, preventing further encouragement of its growth. $\qquad\square$

**Proposition 2.** Our damping loss does not encourage the continuous increase of the damping neuron's softmax value $F_\theta(x)[N+1]$. Once the damping neuron's softmax $F_\theta(x)[N+1] \geq \frac{\lambda}{1+\lambda}$, the gradient of our damping loss $g \geq 0$.

**Proof sketch.** Similar to the proof of Proposition 1, we analyze the gradients from the two components of the damping loss. The damping neuron receives a gradient from the cross-entropy component as $F_\theta(x)[N+1]$ and from the damping component as $\lambda(F_\theta(x)[N+1] - 1)$. Hence, the total gradient is $(1+\lambda)F_\theta(x)[N+1] - \lambda$. Once it reaches $\frac{\lambda}{1+\lambda}$, no further gradient encourages its growth. $\qquad\square$

### 3.4 POWER-SQRT LOSS

Equation 2 details our loss function for each classifier. In an early-exiting architecture, which incorporates multiple classifiers, the total loss function is presented as follows:

$$\min_\theta \sum_{k=1}^K \mathbb{E}_{p(x)} \left[ -\log F_\theta^{(k)}(x)[y] \right] + \sum_{k=1}^K \lambda^{(k)} \mathbb{E}_{p(x)} \left[ -\log F_\theta^{(k)}(x)[N+1] \right] \tag{3}$$

where $k$ presents the index of the classifier. Our damping loss requires different hyperparameters for each classifier, as their optimal values vary, making fine-tuning a challenging task. We found that uniformly setting the hyperparameters $\lambda^{(k)}$ for all classifiers does not yield good performance. While some classifiers improved, others declined. Manually adjusting $\lambda^{(k)}$ for each classifier can enhance performance, but the vast number of possible combinations complicates tuning.

We observed that the values of damping neurons, specifically the damping neuron post-Softmax, are consistently higher in deeper classifiers. A detailed analysis is provided in the Experiment section. This indicates that deeper classifiers are more likely to trigger the damping mechanism, thereby freeing up resources for other classifiers. Additionally, we noted that a higher $\lambda$ value correlates with reduced accuracy in shallow classifiers. Hence, assigning the same $\lambda$ value to shallow classifiers as to deeper ones adversely affects their performance.

Moreover, the damping mechanism should consider all classifiers collectively. Specifically, when some classifiers perform better than others, the damping mechanism should prioritize these, rather than overly diminishing the gradients of underperforming classifiers. This strategy reallocates resources to underperforming classifiers, optimizing overall performance. Thus, a joint-damping mechanism is essential for training classifiers effectively.

Motivated by these findings, we introduce the power-sqrt loss, which dynamically adjusts the hyperparameter $\lambda$ for each classifier, as shown in Figure 2b. We modify the second term of Equation 3 as follows:

$$\lambda \mathbb{E}_{p(x)} \sqrt{\sum_{k=1}^K (-\log F_\theta^k(x)[N+1])^2}. \tag{4}$$

We power the loss associated with the damping neuron of each classifier, sum these squared values, and then extract the square root of the aggregate to form the final loss component.

The power operation intensifies the gradient, focusing it more on the larger values. Since our damping loss must balance the damping component with the cross-entropy component, we apply a square root to the combined damping components across classifiers after performing the power operation. This ensures that the overall damping component remains balanced with the cross-entropy component.

As presented in Proposition 1, the gradient of our damping component generates a stronger inhibitory effect when the softmax value of the correct label is large. After applying the power-sqrt modification, this inhibitory gradient becomes:

$$\frac{-\log F_\theta^k(x)[N+1](F_\theta^k(x)[y])}{\sqrt{\sum_{k=1}^K \left(-\log F_\theta^k(x)[N+1]-1\right)^2}}$$

We present the derivation in the Appendix B.2.

Compared with damping loss, our power-sqrt loss introduces a weight $\frac{-\log F_\theta^k(x)[N+1]}{\sqrt{\sum_{k=1}^K \left(-\log F_\theta^k(x)[N+1]-1\right)^2}}$ to modify the gradient. For different classifiers, denominator of the weight remains the same, while the numerator assigns larger gradients to neurons with smaller damping neuron softmax values.

Since the softmax value of the damping neuron is computed alongside all other neurons that correspond to class labels, a larger softmax value for the correct label is often accompanied by a smaller softmax value for the damping neuron. As a result, our power-sqrt loss focuses the damping gradients more on well-performing classifiers, allowing more parameter space to be allocated to underperforming classifiers.

### 3.5 SAMPLE WISE DYNAMIC TRAINING

The current approach to training early-exiting networks is linear scalarization, where weights are applied to the gradients of different classifiers during training to manage the tradeoffs between them. Our method, however, focuses on evaluating whether these gradients are necessary. After identifying the essential gradients using our approach, linear scaling can still be applied to manage the tradeoff among these necessary gradients.

In contrast to prior meta-learning methods Han et al. (2022); Sun et al. (2022) that learn a meta-net during training to assign classifier weights, our approach derives the weights directly from damping-neuron values, eliminating the meta-net and its training/hyperparameter burden.

Following Han et al. (2022), we normalize the damping-neuron outputs to weights $\tilde{w} \in [-\alpha, \alpha]$ (with $\alpha = 0.8$ on CIFAR and $\alpha = 0.3$ on ImageNet) and set $w = \tilde{w} + 1$. Specifically, for a sample $x_i$, we compute $\tilde{w}_i = -\log F_\theta(x_i)[N+1]$. The classifier-training loss is then $\mathcal{L} = \sum_{k=1}^K \frac{1}{S} \sum_{i=1}^S w_i^{(k)} \mathcal{L}_i^{(k)}$, where $K$ is the number of classifiers, $S$ is the number of samples, and $\mathcal{L}_i^{(k)}$ denotes our power-sqrt loss.

We treat the damping neuron values as constant weights when calculating the gradient of the loss function. This is because incorporating gradients of these weights into the loss gradient would alter the relative importance of different classifiers during training. Specifically, deeper classifiers, which tend to achieve better results, would experience reduced loss. Consequently, if these weights were also differentiated during the gradient computation, instead of being treated as constants, it would encourage increasing the weights applied to deeper networks, thus further reducing the overall loss. However, this would encourage a decrease in the damping neuron values of deeper classifiers, conflicting with the design of our damping loss and degrading performance. Further detailed analyses are provided in the ablation study section.

## 4 EXPERIMENTS

In this section, we evaluate our method through extensive experiments conducted on the CIFAR Krizhevsky et al. (2009) and ImageNet Deng et al. (2009) datasets. Our training strategy is implemented on MSDNet Huang et al. (2017) and RANet Yang et al. (2020), which are representative early-exiting architectures commonly used as backbones to evaluate the performance of related methods Han et al. (2022); Meronen et al. (2024); Gong et al. (2024).

We compare our method with the meta-learning training approach WPN Han et al. (2022) and the feature partitioning method DFS Gong et al. (2024). Furthermore, our method can be integrated with linear scalarization techniques to further enhance performance.

**Datasets.** CIFAR-10 and CIFAR-100 Krizhevsky et al. (2009) both contain 50,000 training images and 10,000 test images. The size of the image is $32 \times 32$. CIFAR-10 has 10 classes, and CIFAR-100 has 100 classes for the classification task. ImageNet Deng et al. (2009) has 1.2 million $224 \times 224$

Table 1: **Anytime prediction results** of a 7-exit MSDNet on CIFAR100.

| Exit | 1 | 2 | 3 | 4 | 5 | 6 | 7 | Avg |
|---|---|---|---|---|---|---|---|---|
| Params ($\times 10^6$) | 0.30 | 0.65 | 1.11 | 1.73 | 2.38 | 3.05 | 4.00 | – |
| FLOPs ($\times 10^6$) | 6.86 | 14.35 | 27.29 | 48.45 | 76.43 | 108.90 | 137.30 | – |
| MSDNet | $61.826 \pm 0.675$ | $64.922 \pm 0.620$ | $67.998 \pm 0.505$ | $71.212 \pm 0.320$ | $73.600 \pm 0.595$ | $75.316 \pm 0.425$ | $75.874 \pm 0.545$ | $70.107 \pm 0.144$ |
| WPN | $62.344 \pm 0.315$ | $65.172 \pm 0.730$ | $68.246 \pm 0.570$ | $71.232 \pm 0.710$ | $73.284 \pm 0.340$ | $74.702 \pm 0.390$ | $74.934 \pm 0.860$ | $69.988 \pm 0.254$ |
| Damping | $61.650 \pm 0.495$ | $64.908 \pm 0.330$ | $68.022 \pm 0.480$ | $71.160 \pm 0.510$ | $73.592 \pm 0.480$ | $75.164 \pm 0.630$ | $75.840 \pm 0.395$ | $70.048 \pm 0.196$ |
| + Power-sqrt | $62.094 \pm 0.295$ | $65.226 \pm 0.420$ | $68.448 \pm 0.405$ | $71.654 \pm 0.460$ | $73.828 \pm 0.295$ | $75.494 \pm 0.375$ | $76.012 \pm 0.470$ | $70.394 \pm 0.215$ |
| + Dynamic | $63.402 \pm 0.330$ | $66.276 \pm 0.345$ | $70.192 \pm 0.395$ | $72.498 \pm 0.300$ | $74.654 \pm 0.490$ | $75.720 \pm 0.385$ | $76.050 \pm 0.475$ | $71.256 \pm 0.149$ |

images for training, 50,000 images for validation and 1000 classes for the classification task. For the sake of fair comparison, we followed Han et al. (2022) setting data augmentations which contain data normalization, random crop, and random flip.

**Backbone architecture and implementation.** Our method can be easily applied to any early exit network. We conduct experiments on two representative early exit architectures, MSDNet Huang et al. (2017) and RANet Yang et al. (2020).

We follow Han et al. (2022) in selecting MSDNet and RANet as backbone architectures. For the CIFAR-100 and CIFAR-10 datasets, we train for 300 epochs with a batch size of 64, using SGD optimizer with a momentum of 0.9 and an initial learning rate of 0.1 decaying with a cosine shape. For the ImageNet dataset, we train for 100 epochs with a batch size of 256, using the same SGD optimizer configuration.

### 4.1 PERFORMANCE EVALUATION

**Results on CIFAR dataset.** We evaluate our training strategy on MSDNet with seven exits, for both the CIFAR-10/100 datasets. We set the hyperparameter $\lambda$ to 0.005 for CIFAR-100 and to 0.075 for CIFAR-10. Initially, we present the 'Anytime Prediction' setting on a 7-exit MSDNet, which details the accuracy of each classifier alongside the corresponding FLOPs (floating point operations, a common metric for assessing the computational budget of the model) as shown in Table 1.

Compared to the MSDNet baseline, our method achieves notable improvements across nearly all classifiers. Additionally, it outperforms the current state-of-the-art meta-learning approach, demonstrating its effectiveness in early-exiting networks.

We show that using the same hyperparameters for all classifiers with our damping loss method led to performance improvements in some classifiers while causing declines in others. However, when incorporating our power-sqrt gradient adjustment, the results showed significant overall improvement. Furthermore, our method is also compatible with existing linear scalarization approaches. When combined with our proposed simplified Dynamic training, it achieves further performance gains.

We also provide the results on the CIFAR-10 dataset in Appendix C.1, demonstrating that our method achieves excellent performance on this dataset as well.

**Results on RANet.** We also conduct experiments on RANet, where our method achieves similarly strong performance. The results and corresponding analysis are provided in Appendix C.3.

**Comparison with label smoothing.** We also compare our method with conventional label smoothing, demonstrating its advantages on early-exiting architectures. Further details are provided in the Appendix C.5.

**Results on ImageNet.** To further demonstrate the effectiveness of our method, we conducted experiments on the large-scale ImageNet dataset. We use MSDNet with five exits as the backbone architecture. For the ImageNet dataset, we set the hyperparameter $\lambda$ to 0.01. Results presented in Table 2 show that our method continues to achieve significant improvements on the large-scale ImageNet dataset.

Table 2: **Anytime prediction results** of a 5-exit MSDNet on ImageNet.

| Exit | 1 | 2 | 3 | 4 | 5 | $\Delta$ |
|---|---|---|---|---|---|---|
| Params ($\times 10^6$) | 4.24 | 8.77 | 13.07 | 16.75 | 23.96 | / |
| FLOPs ($\times 10^9$) | 0.34 | 0.69 | 1.01 | 1.25 | 1.36 | / |
| MSDNet | 59.03 | 66.49 | 70.56 | 72.39 | 74.20 | – |
| WPN | 59.54 | 67.22 | 71.03 | 72.33 | 73.93 | ↑ **1.39** |
| DFS | 61.80 | 68.03 | 70.75 | 71.79 | 72.88 | ↑ **2.58** |
| Power-sqrt | 59.43 | 67.12 | 71.21 | 72.91 | 74.45 | ↑ **2.46** |
| + Dynamic | 59.58 | 67.46 | 71.33 | 73.19 | 74.74 | ↑ **3.63** |

**Dynamic inference results.** In the Dynamic Inference experimental setting, we evaluate a 5-exit MSDNet on the ImageNet dataset, where early-exiting networks dynamically select classifiers based on the computation budget to process incoming data. The anytime prediction results are presented in Table 2. As shown in Fig. 4, deeper classifiers have a greater impact on overall performance in this setting. For instance, the meta-learning approach significantly outperforms the MSDNet baseline in the first three classifiers. However, MSDNet achieves better performance in its deepest classifier. As a result, the performance gap between MSDNet and the meta-learning approach is relatively small under Dynamic Inference.

This is because, in the Dynamic Inference setting, when shallow classifiers misclassify samples, the model has the flexibility to defer the decision to deeper classifiers. Since the performance of the deepest classifier often represents the upper bound of the model's capacity, this setting inherently mitigates the limitations of weaker classifiers. Our method, by filtering out unnecessary gradients, provides the model with greater learning capacity, leading to improved performance in deeper classifiers. Consequently, it achieves competitive results in the Dynamic Inference setting.

## 4.2 ABLATION STUDY

We conduct ablation studies on the 7-exit MSDNet which is used in our CIFAR experiments.

**Adaptive Damping Criterion.** We modify MSD-Net by removing all intermediate classifiers, keeping only the final one to evaluate the effectiveness of our damping loss in a standard setting. In this setup, our method dynamically applies damping to a single classifier based on different training data, without managing multiple classifiers.

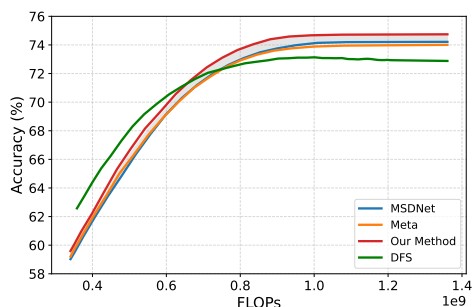

Figure 4: Dynamic inference results on ImageNet.

The results in Table 3 demonstrate the effectiveness of our approach. **MSDNet last only** represents the performance with only the last classifier retained in MSDNet, while **Damping loss last only** applies our damping loss in this setting. We also compare our method with **Threshold last only**, which incorporates a threshold-based gradient selection mechanism into MSDNet with a single classifier.

The results of this experiment confirm that our damping gradients do not degrade classifier performance. Furthermore, as the training states of different data samples vary, our dynamic damping strategy effectively adapts to these variations. Notably, the **MSDNet last only** results presented here are based on the best-performing epoch, following the conventional early stopping method. This highlights that our dynamic damping approach outper-

Table 3: Ablation results on the 7-exit MS-DNet (CIFAR100).

|  | Accuracy |
| --- | --- |
| MSDNet last only | 75.98 |
| Threshold last only | 76.41 |
| Damping loss last only | 76.63 |

forms both Threshold last only and early stopping, demonstrating its superior adaptability in single-classifier training. Moreover, the power-sqrt loss builds on the original damping loss by further utilizing the damping neuron to encode the relative convergence of each classifier, thereby enabling more fine-grained gradient control.

**Sensitivity analysis.** We present the sensitivity analysis of the hyperparameter $\lambda$ in Appendix C.4

Table 4, demonstrating that while our damping loss is relatively sensitive to the choice of $\lambda$, the power-sqrt loss significantly reduces this sensitivity. With the power-sqrt loss, small $\lambda$ values achieve similarly effective results.

**Damping neuron.** We provide a detailed analysis of the damping neuron and its role in dynamic training in Appendix C.2.

## 5 REPRODUCIBILITY STATEMENT

All code, data preprocessing scripts, and training/evaluation pipelines are provided as anonymized supplementary material.

We run experiments on an Nvidia RTX4090 GPU, 12 cores Xeon(R) Platinum 8352V and 90GB RAM. For CIFAR100 experiments, our methods (both damping loss and power-sqrt loss) need approximately 14 hours for total 300 epochs. For ImageNet, our methods need approximately 40 hours for total 100 epochs.

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

# A CONCLUSION

In this paper, we present an adaptive damping mechanism specifically for training early exiting networks. Each classifier's fully connected layer is augmented with a damping neuron, receiving a small gradient to enable adaptive damping when sufficiently trained. Our power-sqrt loss further incorporates a joint consideration of the damping mechanisms across different classifiers. This adaptive damping mechanism significantly enhances the training effectiveness of early exiting networks. By freeing up parameter space that would typically be wasted on overfitting in traditional training methods, the performance of early exiting networks significantly outperforms the current methods. Furthermore, our approach is compatible with state-of-the-art linear scalarization training methodologies.

# B APPENDIX FOR THEORY

## B.1 PROOF OF PROPOSITION 1 AND PROPOSITION 2

We present a detailed proof of the gradients received by each neuron in the fully connected layer from our damping loss.

Our damping loss's gradient has two components: the cross-entropy part $\triangledown - \log F_\theta(x)[y]$ and the gradient of our damping neuron $\triangledown - \log F_\theta(x)[N+1]$. We demonstrate the gradients transmitted from the loss of the damping neuron part to each neuron as follows:

$$\frac{\partial - \log F_\theta(x)[N+1]}{\partial f_\theta(x)[i]} \tag{5}$$

According to the chain rule:

$$= -\frac{1}{F_\theta(x)[N+1]} \times \frac{\partial F_\theta(x)[N+1]}{\partial f_\theta(x)[i]} \tag{6}$$

$$= -\frac{1}{F_\theta(x)[N+1]} \times \frac{\partial \frac{\exp(f_\theta(x)[N+1])}{\sum_{j=1}^{N+1} \exp(f_\theta(x)[j])}}{\partial f_\theta(x)[i]} \tag{7}$$

According to the quotient rule:

$$= -\frac{1}{F_\theta(x)[N+1]} \times \left(\frac{\partial \exp(f_\theta(x)[N+1])}{\partial f_\theta(x)[i]} \times \frac{1}{\sum_{j=1}^{N+1} \exp(f_\theta(x)[j])}\right. \tag{8}$$

$$\left. - \exp(f_\theta(x)[N+1]) \times \frac{\partial \sum_{j=1}^{N+1} \exp(f_\theta(x)[j])}{\partial f_\theta(x)[i]} \times \frac{1}{\left(\sum_{j=1}^{N+1} \exp(f_\theta(x)[j])\right)^2}\right)$$

$$= -\frac{1}{F_\theta(x)[N+1]} \times \left(\frac{\partial \exp(f_\theta(x)[N+1])}{\partial f_\theta(x)[i]} \times \frac{1}{\sum_{j=1}^{N+1} \exp(f_\theta(x)[j])}\right. \tag{9}$$

$$\left. - \exp(f_\theta(x)[N+1]) \times \frac{\partial \exp(f_\theta(x)[i])}{\partial f_\theta(x)[i]} \times \frac{1}{\left(\sum_{j=1}^{N+1} \exp(f_\theta(x)[j])\right)^2}\right)$$

$$= -\frac{1}{F_\theta(x)[N+1]} \times \left(\frac{\partial \exp(f_\theta(x)[N+1])}{\partial f_\theta(x)[i]} \times \frac{1}{\sum_{j=1}^{N+1} \exp(f_\theta(x)[j])}\right. \tag{10}$$

$$\left. - \exp(f_\theta(x)[N+1]) \times \exp(f_\theta(x)[i]) \times \frac{1}{\left(\sum_{j=1}^{N+1} \exp(f_\theta(x)[j])\right)^2}\right)$$

$$= -\frac{1}{F_\theta(x)[N+1]} \times \left(\frac{\partial \exp\left(f_\theta(x)[N+1]\right)}{\partial f_\theta(x)[i]} \times \frac{1}{\sum_{j=1}^{N+1} \exp\left(f_\theta(x)[j]\right)}\right. \tag{11}$$
$$\left. - F_\theta(x)[N+1] \times F_\theta(x)[i]\right)$$

when $i \neq N+1$, $\frac{\partial \exp(f_\theta(x)[N+1])}{\partial f_\theta(x)[i]} = 0$,

$$eq.(7) = -\frac{1}{F_\theta(x)[N+1]} \times -F_\theta(x)[N+1] \times F_\theta(x)[i]$$

$$= F_\theta(x)[i]$$

when $i = N+1$, $\frac{\partial \exp(f_\theta(x)[N+1])}{\partial f_\theta(x)[i]} = \exp\left(f_\theta(x)[N+1]\right)$,

$$eq.(7) = -\frac{1}{F_\theta(x)[N+1]} \times \left(F_\theta(x)[N+1] - (F_\theta(x)[N+1])^2\right)$$

$$= F_\theta(x)[N+1] - 1$$

Thus, for the damping neuron, where $i = N+1$, the gradient from damping component is $F_\theta(x)[N+1] - 1$. For all other neurons, the gradient is $F_\theta(x)[i]$.

The same derivation applies to the gradient of the cross-entropy component as well.

The gradients from the cross-entropy component are $F_\theta(x)[y] - 1$ for the neuron corresponding to the correct label, and $F_\theta(x)[i]$ for the other neurons.

### B.2 DERIVATION

The damping component gradient for the neuron corresponding the correct label from power-sqrt loss is:

$$\frac{\partial \sqrt{\sum_{k=1}^{K}(-\log F_\theta^k(x)[N+1])^2}}{f_\theta(x)[y]}.$$

Same with the Proof above, the $k-$th classifier's gradient from damping component for the neuron corresponding the correct label is: $F_\theta^k(x)[y])$.

According to the chain rule, we get final gradient: $\dfrac{-\log F_\theta^k(x)[N+1](F_\theta^k(x)[y])}{\sqrt{\sum_{k=1}^{K}\left(-\log F_\theta^k(x)[N+1]-1\right)^2}}$

## C  APPENDIX FOR EXPERIMENT

### C.1  RESULT ON CIFAR10

We also present results on the CIFAR10 dataset in Table 1, where our method continues to achieve stable improvements. Notably, the enhancements on CIFAR10 are less pronounced compared to those on CIFAR100. This discrepancy arises because we employ the same MSDNet model architecture for both CIFAR100 and CIFAR10, and the parameter space provided by the model is more than sufficient for CIFAR10, thus reducing the impact of overfitting. This comparison further illustrates the effectiveness of our method in unlocking the potential of the parameter space.

### C.2  DAMPING NEURON.

In this section, we conduct a detailed analysis of the values generated by the damping neuron and their effects, as well as their gradients in dynamic training. We keep the hyperparameter $\lambda$ to 0.005 for damping loss, power-sqrt, and dynamic training.

Table 4: **Anytime prediction results** of a 7-exit MSDNet on CIFAR10

| Exit index | 1 | 2 | 3 | 4 | 5 | 6 | 7 |
|---|---|---|---|---|---|---|---|
| Params($\times 10^6$) | 0.30 | 0.65 | 1.11 | 1.73 | 2.38 | 3.05 | 4.00 |
| FLOPs($\times 10^6$) | 6.86 | 14.35 | 27.29 | 48.45 | 76.43 | 108.90 | 137.30 |
| MSDNet | 88.51 | 90.38 | 92.15 | 93.21 | 93.89 | 94.22 | 94.54 |
| Meta-learning Early Exiting | 88.54 | 90.19 | 91.61 | 92.55 | 93.28 | 93.40 | 93.67 |
| + Power-sqrt | 88.38 | 90.33 | 92.06 | 93.71 | 94.23 | 94.45 | 94.49 |
| + Dynamic training | **88.42** | **90.21** | **92.01** | **93.58** | **94.27** | **94.50** | **94.55** |

Table 5 provides a detailed presentation of the average values of the damping neuron for each classifier across different methods on the test set. In Table 5, the numbers on the left represent accuracy, while the bolded values in parentheses indicate the average values of the damping neuron. We observe that the values of the damping neuron are consistently higher in deeper classifiers compared to shallower ones. This suggests that deeper classifiers are more likely to achieve superior training performance, thereby more frequently activating the damping mechanism. We observe that while deeper classifiers tend to have higher damping neuron values, these values are smaller in the power-sqrt version of the damping neuron. This occurs because the power-sqrt approach takes into account the training conditions of different classifiers collectively.

We delve deeper into the analysis of weight gradients in dynamic training. As seen from Table 5, when these weights possess gradients during training, they tend to assign larger weights to deeper classifiers since they exhibit lower losses, thereby aiming for an overall reduction in total loss. However, this conflicts with the design of our damping loss. It is observed that, when retaining the gradients of weights, deeper networks paradoxically exhibit smaller damping neuron values, contrary to the previously observed pattern. This conflict can lead to a decline in training performance. When we use the values of the damping neurons as weights without propagating gradients, the outcomes are generally consistent with our earlier observations of our power-sqrt version and yield better performance.

Table 5: **Ablation study of damping neuron.** The bolded values in parentheses present the average values of damping neurons

| Exit index | 1 | 2 | 3 | 4 | 5 | 6 | 7 |
|---|---|---|---|---|---|---|---|
| Params($\times 10^6$) | 0.30 | 0.65 | 1.11 | 1.73 | 2.38 | 3.05 | 4.00 |
| FLOPs($\times 10^6$) | 6.86 | 14.35 | 27.29 | 48.45 | 76.43 | 108.90 | 137.30 |
| Damping loss | 61.53(**0.006**) | 64.71(**0.008**) | 68.22(**0.015**) | 71.27(**0.028**) | 73.76(**0.050**) | 75.27(**0.070**) | 75.76(**0.071**) |
| + Power-sqrt | 62.07(**0.002**) | 65.44(**0.003**) | 69.32(**0.006**) | 71.61(**0.012**) | 73.88(**0.024**) | 75.89(**0.036**) | 76.45(**0.037**) |
| + Dynamic training | 62.74(**0.004**) | 65.69(**0.006**) | 69.76(**0.010**) | 71.77(**0.019**) | 74.61(**0.031**) | 75.96(**0.049**) | 76.63(**0.044**) |
| Dynamic training with gradient | 62.66(**0.041**) | 66.19(**0.029**) | 70.19(**0.020**) | 72.13(**0.010**) | 74.28(**0.001**) | 75.64(**0.004**) | 76.07(**0.006**) |

## C.3 RESULTS ON RANET.

We extended our experiments to include RANet, another representative early exiting architecture, to demonstrate the generality of our method. The anytime prediction results for the six exit RANet are displayed in Table 6. The improvements observed with our method on RANet are more significant than those on MSDNet.

This difference can be attributed to RANet's hierarchical processing of data resolutions, where shallow classifiers operate on low-resolution data, while deeper classifiers are exclusively fed high-resolution features. This design increases the disparity between features processed at different classifier depths, making the negative impact of classifier overfitting on others more severe. Consequently, the reduction of unnecessary gradients and the optimized parameter space utilization provided by our method become even more crucial in mitigating this effect.

Table 6: **Anytime prediction results** of a 6-exit RANet on CIFAR100

| Exit index | 1 | 2 | 3 | 4 | 5 | 6 |
|---|---|---|---|---|---|---|
| Params($\times10^6$) | 0.36 | 0.90 | 1.30 | 1.80 | 2.19 | 2.62 |
| FLOPs($\times10^6$) | 8.37 | 21.79 | 32.88 | 41.57 | 53.28 | 58.99 |
| RANet | 65.28 | 68.16 | 70.52 | 70.64 | 72.39 | 72.75 |
| WPN | 65.33 | 68.69 | 70.36 | 70.80 | 72.57 | 72.45 |
| Ours | 65.63 | 68.96 | 71.49 | 71.65 | 73.19 | 73.69 |
| + Dynamic training | 65.67 | 69.38 | 71.88 | 71.92 | 74.19 | 74.26 |

Table 7: **Ablation study of hyperparameter $\lambda$** *(values are mean $\pm$ std)*

| Method | C1 | C2 | C3 | C4 | C5 | C6 | C7 | Avg |
|---|---|---|---|---|---|---|---|---|
| damping 0.005 | $61.650 \pm 0.495$ | $64.908 \pm 0.330$ | $68.022 \pm 0.480$ | $71.160 \pm 0.510$ | $73.592 \pm 0.480$ | $75.164 \pm 0.630$ | $75.840 \pm 0.395$ | $70.048 \pm 0.196$ |
| damping 0.025 | $61.360 \pm 0.440$ | $64.802 \pm 0.395$ | $67.792 \pm 0.555$ | $70.782 \pm 0.255$ | $73.746 \pm 0.710$ | $75.640 \pm 0.800$ | $76.238 \pm 0.630$ | $70.051 \pm 0.212$ |
| damping 0.05 | $61.096 \pm 0.545$ | $64.136 \pm 0.825$ | $67.872 \pm 0.610$ | $70.652 \pm 0.325$ | $73.492 \pm 0.385$ | $75.510 \pm 0.350$ | $75.750 \pm 0.470$ | $69.787 \pm 0.248$ |
| damping 0.075 | $61.408 \pm 0.735$ | $64.378 \pm 0.315$ | $68.036 \pm 0.745$ | $70.756 \pm 0.145$ | $73.566 \pm 0.145$ | $75.344 \pm 0.190$ | $75.818 \pm 0.360$ | $69.901 \pm 0.129$ |
| power-sqrt 0.005 | $62.094 \pm 0.295$ | $65.226 \pm 0.420$ | $68.448 \pm 0.405$ | $71.654 \pm 0.460$ | $73.828 \pm 0.295$ | $75.494 \pm 0.375$ | $76.012 \pm 0.470$ | $70.394 \pm 0.215$ |
| power-sqrt 0.025 | $61.884 \pm 0.480$ | $65.432 \pm 0.525$ | $68.578 \pm 0.425$ | $71.788 \pm 0.670$ | $74.152 \pm 0.160$ | $75.638 \pm 0.135$ | $76.116 \pm 0.400$ | $70.513 \pm 0.139$ |
| power-sqrt 0.05 | $62.010 \pm 0.405$ | $65.046 \pm 0.305$ | $68.594 \pm 0.470$ | $71.560 \pm 0.565$ | $74.028 \pm 0.320$ | $75.860 \pm 0.395$ | $76.238 \pm 0.535$ | $70.477 \pm 0.086$ |
| power-sqrt 0.075 | $61.806 \pm 0.360$ | $65.028 \pm 0.450$ | $68.520 \pm 0.940$ | $71.708 \pm 0.335$ | $74.192 \pm 0.535$ | $75.996 \pm 0.195$ | $76.220 \pm 0.255$ | $70.496 \pm 0.221$ |

Table 8: **Comparison with label smoothing** on 7-exits MSDNet on CIFAR100.

| Exit | 1 | 2 | 3 | 4 | 5 | 6 | 7 |
|---|---|---|---|---|---|---|---|
| MSDNet | 60.78 | 64.54 | 68.51 | 71.41 | 73.68 | 75.61 | 76.31 |
| Label smoothing | 61.33 | 64.80 | 68.32 | 70.88 | 73.10 | 74.75 | 75.68 |
| **Power-sqrt** | **62.07** | **65.44** | **69.32** | **71.61** | **73.88** | **75.89** | **76.45** |

## C.4 SENSITIVITY ANALYSIS.

The objective of training early-exiting models is to jointly optimize the performance of all classifiers. To evaluate the stability of our method under different hyperparameters, we conduct five runs with different random seeds for each value of the damping weight $\lambda$. We report the accuracy of each classifier as well as the overall average performance. Results are shown in Table 4

## C.5 COMPARISON WITH LABEL SMOOTHING

While our method may appear superficially similar to confidence-based regularization techniques such as label smoothing, its design and purpose are fundamentally different. Label smoothing applies a uniform confidence penalty to each classifier independently. By contrast, our method is specifically tailored for early-exiting architectures, where all classifiers must be trained jointly.

To achieve this, we introduce a *damping neuron* whose output acts as a learned coordination signal. This signal drives two key components of our approach:

- **Dynamic training**, where it adaptively controls the weight assigned to each classifier's gradient;
- **Power-sqrt loss**, where it regulates the degree of gradient damping across classifiers.

The critical distinction is that our method explicitly coordinates the optimization of all classifiers through a shared and interpretable mechanism, rather than regularizing them in isolation.

To further highlight this difference, we compared our approach with label smoothing in early-exiting experiments (Table 7). These experiments are conducted on well-established image classification benchmarks (e.g., MSDNet backbones), which already incorporate mature overfitting-prevention techniques. In this setting, label smoothing fails to provide additional benefits. In contrast, our method directly addresses the unique challenge of early-exiting training—*jointly optimizing multiple classifiers under potential gradient conflicts*—and thus yields consistent improvements.

