# OpenReview forum: "Diminishing Non-important Gradients for Training Dynamic Early-Exiting Networks"
_ICLR.cc/2026/Conference — Submitted to ICLR 2026_

### Official Review · Reviewer_FsJU · 2025-10-30

**Soundness:** 2
**Presentation:** 2
**Contribution:** 2
**Rating:** 4
**Confidence:** 4

**Summary:**

This submission studies the training of multi-exit dynamic networks. It reformulates a K-class classification problem to a (K+1)-class one by adding a separate neuron at each linear classifier. It is claimed that this additional class can prevent the useless/harmful gradients for those samples that achieve a high softmax score. Experiments with a classic multi-exit model, MSDNet, on CIFAR and ImageNet validate that the proposed method improves the performance of multi-exit models.

**Strengths:**

1. The motivation is reasonable and clear;

2. Experiments in section 3.2 (Fig. 3) successfully support the authors' claim that the gradients of those samples with high softmax scores might be useless or harmful.

**Weaknesses:**

1. The logic of the proposed method is confusing and not explained clearly.
    - I understand that the second item (damping loss) in Eq. (2) can dominate when a sample achieves a high softmax score (the first item is low). However, until Eq. (2) I can get that the intrinsic operation is setting the additional class also as a "ground-truth" label. This intrinsic logic is not clearly stated in the introduction, Figs. 1 & 2.
    - The relationship between damping loss and the proposed power-sqrt loss is confusing. Eq. (3) is the extension of Eq. (2). But I did not fully understand the motivation and essence of Eq. (4). What does the following gradient equation imply?
    - In summary, the underlined logics of the authors' proposed losses and equations are not stated with clear words.

2. I also have some concerns about the experiments.
    - In Tab. 2, The MSDNet baseline's performance is different with the reported results in WPN [1], while the performance of WPN [1] is the same with the reported results in [1]. This raises some concerns about fair comparison.
    - The RANet baseline's performance (Tab. 6) also differs with the reported on in [1] (Tab. 2).
    - The two curves of MSDNet and Meta (WPN?) are close in Figure 4. This is also not aligned with the results in [1] (Fig. 7).
    - I'm also curious about the effectiveness on latest architectures, such as Dyn-Perceiver [2].

3. An important baseline is missed [3]. It also aims at the calibration of gradients in multi-exit models.

[1] Learning to Weight Samples for Dynamic Early-exiting Networks. ECCV, 2022

[2] Dynamic Perceiver for Efficient Visual Recognition. ICCV, 2023

[3] proved techniques for training adaptive deep network. ICCV, 2021.

**Questions:**

Please see the weakness part.

---

> ### Author Response · Authors · 2025-12-04
>
> 1) Why setting the additional class also as a "ground-truth" label
>
>
> We have updated the Introduction and Motivation sections of the paper. Our method aims to jointly consider the training states of different classifiers by observing that gradients become increasingly unnecessary as the softmax confidence grows. Assigning a ground-truth label to the extra neuron allows us to dynamically damp these gradients, as demonstrated in Proposition 1.
>
> 2) Why equation 4
>
>
> Our Eq. 4 introduces the power-sqrt loss. Compared with the damping loss, it concentrates gradient suppression on the classifiers where suppression is more necessary, thereby providing more training flexibility for the other classifiers.

---

### Official Review · Reviewer_uRxk · 2025-10-30

**Soundness:** 2
**Presentation:** 1
**Contribution:** 3
**Rating:** 4
**Confidence:** 2

**Summary:**

To improve the inference efficiency, early-exiting networks have been proposed, which attach multiple intermediate classifiers and terminate the inference once a intermediate classifier meets a predefined confidence threshold. However, in current early-exiting networks training process, unnecessary gradients may be used to update the model parameters, leading to performance degradation. To resolve this issue, this work introduces a damping neuron into the layer before the classifier softmax, which is used to dynamically reduces unnecessary gradients during training. In addition, a new power-sqrt loss is proposed to improve the efficiency of the algorithm. Empirical results demonstrate that the proposed method outperforms other tested baselines on both CIFAR and ImageNet datasets.

**Strengths:**

1. This paper has considered an important problem of early-exiting networks. The ideas of damping neuron and damping loss are new. The proposed architecture can be achieved by simply adding a neuron for each classifier, leading to an easy and efficient implementation.

2. The empirical results are good. The proposed method outperforms other tested baselines in most cases. Besides, this work has conducted a comprehensive ablation study to show the effectiveness of each new design.

**Weaknesses:**

1. The introduction and related work sections of this paper are poorly organized, which makes the overall motivation of the study unclear. Firstly, I don't understand why the paper keeps highlighting the "gradient conflict" issue which seems not related to this work. More importantly, it seems that the original motivation of the proposed method is that "unimportant" gradients may cause the early-exiting model to overfit during training. However, overfitting has been a well-explored issue, and numerous methods have been proposed to address it. This paper does not provide any discussion or comparison of the existing approaches. Also, in the experiments, the proposed method is demonstrated to outperform other tested baselines. However, is the observed performance improvement truly a result of mitigating overfitting? If yes, a comparison between the training loss and validation loss should be shown to verify this. Otherwise, I recommend the authors to reconsider the motivation of the proposed method.

2. The writing of this paper needs to be improved. There are many confusing or inaccurate claims that make the paper hard to follow. For example,

* Throughout this paper, the "unnecessary gradients" are not clearly defined. Does it refer to the gradients that hurt the model performance or the gradients that have a relatively small impact on the model performance? Due to the lack of a formal definition, the correctness and rationality of the theoretical analyses are hard to assess.

* In line 067, it's not clear why a higher gradient of the damping neuron can effectively prevent unnecessary gradients.

* In line 199, it's confusing why "maintaining a higher loss provides additional capacity for other classifiers". The results in Figure 3 do not seem to demonstrate this observation.

* The cross-entropy loss in (2) is wrong.

* The relationship between Proposition 1 and the claims above is not clear. Also, the definition of $g$ is confusing. What does the "gradient from our damping loss" refer to?

* All the theoretical conclusions and the propositions 1&2 are described based on the loss function (2). However, solely using the damping trick designed in (2) cannot achieve desirable results (e.g., as shown in Table 1). The properties of the better-performed loss functions in (4) and line 347 are not comprehensively discussed.

**Questions:**

1. Can the results in propositions 1&2 be generalized to the dynamic training scenario in Section 3.5?

2. Can the proposed method be generalized to the more popular LLMs or other transformer-based structures?

---

> ### Author Response · Authors · 2025-12-04
>
> 1) Paper motivation
>
> We have revised our introduction and related work to make our motivation more clear. Gradients of different layers may conflict, and that conflict is alleviated by damping gradient values that might be unnecessarily high, saving parameter space for improving other layers. Hence, our paper is motivated by alleviating both gradient conflicts and overfiting (as unusually high softmax scores may lead to overfitting).
>
>   However, as mentioned to ojs8, multiple classifiers must be optimised jointly in our setting, and conventional overfitting mitigation techniques are not well suited to this setting.
>
> 2) Notation and Correctness
>
> Our cross-entropy loss (2) is not wrong. We add an additional term to the traditional cross entropy loss in order to include our damping neuron, as described in the paper.
>
>
>   Similarly, $g$ refers to the gradient computed during training using our modified loss function. As explained in the paper, that gradient would be smaller for layers with high soft-max values due to the damping component, allowing the gradient part corresponding to the other layers to have a greater impact.
>
>
>   Regarding Table 1, as explained in the paper, only using the damping neuron would still require fitting of $\lambda$ values for each layer. We solve this issue by using the power-sqrt loss. Hence, the table shows that our full approach provides the best results. Only using the damping neuron (without power-sqrt loss) could provide better results, but it would require tuning of $\lambda$ values.

---

### Official Review · Reviewer_e4Nu · 2025-11-01

**Soundness:** 2
**Presentation:** 2
**Contribution:** 2
**Rating:** 4
**Confidence:** 4

**Summary:**

This manuscript proposes a novel adaptive damping training strategy to address unnecessary gradients in dynamic early-exiting networks. Unlike existing state-of-the-art methods that only balance gradients without evaluating their necessity, the proposed approach adds a damping neuron to the last fully connected layer of each classifier and uses a damping loss to reduce non-beneficial gradients, while introducing a power-sqrt loss to concentrate damping neuron gradients on better-performing classifiers. Extensive experiments on CIFAR-10/100 and ImageNet datasets using MSDNet and RANet backbones show significant accuracy improvements across all classifiers with negligible computational increases, outperforming existing methods.

**Strengths:**

1. this paper focus a important problem —— improving the performance of the promising early exiting dynamic networks.
2. the experiments conducted on both cifar and imagenet, which are competitive benckmarks.
3. the sampling wise dynamic training is interesting, which eliminate the meta-learning scheme in previous works.

**Weaknesses:**

1.  logic of the methods unclear: the core idea of this paper is add an extra neuro in the final layer. However, is the neuro a activated value, or an extra channel in the Linear layer is unclean.
2. although the overall method look new to me, how the finding in sec 3.2 motivate the authors to design the following method is unclear.
3. the bugeted inference results are missing, make the reviewers can only see the results on each exit.

**Questions:**

1. how the proposed method reflect the finding in Sec 3.2 is unclear. In Sec 3.2, it seems the loss of over 95% confident samples should be dropped. How the extra neuro in Sec 3.3 suppress the over confident neuro's gradient?
2. How the extra neruo is supervised? It should be clarified explictly in the method section.

---

> ### Author Response · Authors · 2025-12-03
>
> We thank the reviewer for the helpful comments. We address each point below.
>
>
> 1) Clarification on the Implementation of the Damping Neuron
>
> We add one extra neuron to the final fully connected layer of each classifier. This neuron is identical to the neurons used for class logits. It is simply an additional output channel in the linear layer.
>
>
> 2) Missing Budgeted Inference Results
>
> We already provided the budgeted inference results in Figure 4 of the paper.
>
> 3) Unclear Connection Between Sec. 3.2 Findings and the Damping Neuron Mechanism
>
>
> Existing methods focus on balancing gradients but ignore whether these gradients are necessary.
> Our experiments in Sec. 3.2 show that as the softmax confidence increases, the associated gradients are more likely to be unnecessary. Therefore, our damping loss assigns larger damping gradients to high-confidence samples, allowing us to jointly train the multiple classifiers in early-exiting networks.
>
>
> We have revised the Introduction and Sec. 3.2 to make the motivation clearer.
>
>
>
> 4) Unclear Supervision of the Damping Neuron
>
> The supervision of the damping neuron is defined in Eq. (2), which presents the loss function. Specifically, our damping loss adds a small additional gradient to the extra damping neuron on top of the standard cross-entropy loss.

---

### Official Review · Reviewer_ojs8 · 2025-11-01

**Soundness:** 2
**Presentation:** 2
**Contribution:** 2
**Rating:** 2
**Confidence:** 4

**Summary:**

This paper proposes a new training strategy for early-exit networks through adaptive gradient damping. The method introduces a damping neuron into each early exit classifier’s fully connected layer and a corresponding damping loss that selectively suppresses gradients deemed non-beneficial based on softmax confidence. If the softmax value is high, the gradient for that classifier is dampened more. Additionally, a power-sqrt loss is designed to distribute damping across early exit classifiers in proportion to their relative performance. Experiments on CIFAR-10, CIFAR-100, and ImageNet using MSDNet and RANet show improvements over baselines and meta-learning approaches such as WPN, with minimal computational overhead.

**Strengths:**

The idea of adaptively diminishing “non-important” gradients is conceptually interesting and differentiates this work from prior gradient conflict–resolution approaches.

The proposed approach is complementary to existing multi-task and early-exit training strategies (e.g., linear scalarization, meta-learning).

Results across three datasets and two architectures (MSDNet, RANet) show small but consistent accuracy gains.

The method introduces minimal computational cost (a single neuron per early exit classifier).

**Weaknesses:**

I am not convinced that the main results in Tables 1, 4, and 6 are truly “notable,” “strong,” or “excellent,” as claimed by the authors. The improvements over the baseline are small, and Table 7 shows relatively large standard deviations. This raises the question of whether the differences between the baseline and the proposed method are statistically significant.

The empirical observation that high-softmax-value samples can yield harmful gradients is backed by an experimental result (Fig. 3). However, more details are needed for this experiment, e.g. which dataset? how many classes? is the accuracy in Fig 3 test set accuracy? How about loss?

Reported accuracy gains are modest, especially given the additional loss design complexity.

The mathematical derivation of the “power-sqrt loss” is somewhat contrived and lacks intuitive grounding or theoretical justification.

It remains unclear how the damping neuron behaves dynamically and why the square-root aggregation is the optimal design choice.

Comparisons are restricted mainly to WPN (2022) and DFS (2024). More recent adaptive early-exit or uncertainty-aware training methods should be included for a fair benchmark. Examples:
* Ilhan et al. Adaptive deep neural network inference optimization with EEnet. WACV2024
* Bajpati, Hanawal. BEEM: Boosting performance of early exit DNNs using multi-exit classifiers as experts. ICLR2025
* Lin et al. E3: Early exiting with explainable ai for real-time and accurate dnn inference in edge-cloud systems
* Xu et al. Early exit via class means for efficient supervised and unsupervised learning. IJCNN2022

While several ablations are presented, they do not isolate contributions cleanly. For example, the separate effects of damping loss vs. power-sqrt vs. dynamic training are not statistically validated.

The term "trade-off between gradients" is not clear. The paper would benefit from a clear (preferably mathematical) definition. Related to this, proposed power-sqrt does a global adjustment which could be seen as making a trade-off among gradients.

Some equations and figures are difficult to follow (e.g., Fig. 2b caption), and notation is occasionally inconsistent or verbose. A numerical toy example with two or three classes would be much easier to understand the intuition behind how dampening works.

Most papers are cited using the \citet command. they should have used \citep.

There is paragraph on overfitting in Related Work. I am not sure why this body of work is necessary here.

The text has several grammatical and stylistic issues that obscure key ideas and would benefit from careful editing. Typo at L484: early-existing.

**Questions:**

- Can you clarify the details about Fig 3?
- Are improvements in tables 1,4 and 6 statistically significant?
- How does your accuracy & computational savings compare to those of other early-exit methods that I mention above?

---

> ### Author Response · Authors · 2025-12-04
>
> 1) Statistical significance of the experiment results
>
>
> We reran all methods in Table 1 with five different random seeds. We have updated Table 1 to present these results.
>
>
> The results show that our method achieves significant improvements over the baseline. Notably, training early-exiting networks requires balancing the performance of multiple classifiers, where improving one classifier often degrades another. In contrast, our method yields clear and consistent gains across all classifiers.
> The results also clearly show that the improvements brought by our different modules are significant.
>
>
>
> 2) Experiment details of Fig.3
>
> In Fig. 3, we use the same experimental setup as in Table 1 but remove the intermediate classifiers and train only the final classifier to more clearly illustrate the effect. The accuracy is measured on the test set, and the loss refers to the training loss. We have revised this part of the paper to make it clearer.
>
> 3) Why overfitting in related work
>
> Our work focuses on the necessity of different gradients during the training of early-exiting networks, which is related to but different from standard overfitting issues. In early-exiting architectures, multiple classifiers must be optimized jointly, and conventional overfitting mitigation techniques are not well suited to this setting.
>
> 4) Typos
>
>
> Thank you for the detailed suggestions. We have corrected these typos in the paper.

---

### Meta-Review · Area_Chair_2Pbh · 2026-01-06

**Summary:**

Reviewer ojs8 has raised several weaknesses, including

- Not convinced that the main results in Tables 1, 4, and 6 are truly “notable,” “strong,” or “excellent,” as claimed by the authors. The improvements over the baseline are small, and Table 7 shows relatively large standard deviations.

-- Among them, although the authors reran the results of Table 1, it is still questionable whether the claims are justified.

- Comparisons are restricted mainly to WPN (2022) and DFS (2024). More recent adaptive early-exit or uncertainty-aware training methods should be included for a fair benchmark. Examples: Ilhan et al, WACV2024; Bajpati, Hanawal, ICLR2025; Lin et al. E3: SenSys'25; Xu et al, IJCNN2022.

- The mathematical derivation of the “power-sqrt loss” is somewhat contrived and lacks intuitive grounding or theoretical justification,

Reviewer FsJU has pointed out some concerns about the experiments and an important baseline is missed.

At least for these issues, I don't consider them to be well addressed or revised.

**Reviewer Concerns:**

In addition to the above, all reviewers have the comments on the quality of writing of this paper, including "Some equations and figures are difficult to follow, and notation is occasionally inconsistent or verbose," "logic of the methods unclear," and "The introduction and related work sections of this paper are poorly organized."

The authors have only partly addressed the concerns. They revised the Introduction, Sec. 3.2, and the Introduction and Motivation sections, yet without providing a sufficiently clear description of how the paper has been revised or improved.

**Reviewer Scores:**

I think most reviewers tend to maintain or lower the scores.

---

### Decision · Program_Chairs · 2026-01-26

Reject